# The Relationship between Strategic Orientation, Service Innovation, and Performance in Hotels in Angola

**Gerdina Handa Serafim and José Manuel Cristóvão Veríssimo \***

ISEG—Lisbon School of Economics & Management, Universidade de Lisboa, 1200-781 Lisbon, Portugal; gerdinahanda@phd.iseg.ulisboa.pt
* Correspondence: jose.verissimo@iseg.ulisboa.pt

**Abstract:** This paper aims to investigate the impacts of customer orientation, competitor orientation, learning orientation, technology orientation, and entrepreneurial orientation on hotel innovation and performance. Data from 69 hotels in four Angolan provinces were analyzed using the partial least squares (PLS) approach and multi group analysis. The results show that learning and entrepreneurial orientations have a positive impact on hotel innovation. As anticipated, innovation has a positive impact on performance. According to the multigroup analysis, only the hotel category has a moderating effect on performance. Results suggest that hotels in developing countries could add value to both customers and shareholders by promoting new services and exploring new business opportunities. To the best of our knowledge, this is one of the few studies that has researched the impact of strategic orientation on hotel innovation and financial performance in developing countries.

**Keywords:** service innovation; customer orientation; competitor orientation; learning orientation; technological orientation; entrepreneurship orientation; hospitality

## 1. Introduction

Competition and the explosion of globalised technological innovation and differentiation have come to be considered as requirements for any company [1]. The development of innovations requires a strategic posture, the support of the organisational structure and the existence of administrative processes that can adapt to uncertain environments [2]. This is even more demanding in developing countries due to the shortage of skilled resources. This research follows the extant literature and considers that innovation in the service sector depends on customer orientation, competitor orientation, organisational learning orientation, entrepreneurship orientation, and technological orientation [3,4], where dependable competencies need to be developed to drive good organisational performance, be they financial or non-financial factors, which correspond to a strategic orientation [5–11].

Customer orientation represents the degree to which a company obtains and uses customer information and uses it to develop a strategy that efficiently meets customers' needs and desires [1,12]. Customer orientation does not necessarily mean listening to customers, as customers can influence the imposition of strict limits on the strategies that companies may or may not follow [13,14], which, when related to innovation, can have a positive impact on a company's performance [15–18].

Competitor orientation involves the ability to create value to improve a company's performance by looking at competitors [19] and trying to anticipate trends and demands [20]. Companies tend to perform better when they face very competitive competitors, albeit this depends on their response capacity [21], as in highly competitive environments knowing and understanding the competition enables the company to survive with success [22,23], rather than lose customers and consequently market share. On the one hand, learning orientation is a set of organisational values which affect the creation of value and the implementation of knowledge [2], which leads to the proactive adoption of a business

strategy and renders the organisation more innovative [24], which in turn positively affects performance [25]. On the other hand, technology orientation promotes openness to ideas that use cutting-edge technologies and is proactive with regards the acquisition and integration of new and sophisticated technologies in the process of developing new products [25], which thus facilitates innovative performance [26–29] as well as the company's performance [30,31]. Finally, entrepreneurial orientation results in the development of activities, which leads to the design of new products and/or different services which improve efficiency, cost reduction, value creation, and customer loyalty [5,32], and consequently influence innovation [12,33], which implies a positive implication on the company's performance [9,34,35]. To the best of the authors' knowledge, this is one of the few papers that examines the impact of strategic orientation on innovation and financial performance in hotels in developing countries, which not only help develop the domestic economy by creating jobs in the tourism industry, but also increase regional awareness.

## 2. Theoretical Framework and the Development of Hypotheses

### 2.1. Strategic Guidelines, Innovation, and Performance

The impact of strategic orientations on hotel innovation corresponds to the adoption of a set of strategies oriented to the activities of the hotel unit [36], through the production of behaviours that guarantee not only a good performance, but also the hotel's survival [28,35]. The strategic orientations adopted by hotels have a significant impact on the innovations to be introduced in the market [28], as they are related to the adoption of a set of principles that direct and influence the activities developed in a determined hotel unit [37] and thus generate behaviours that ensure viability and performance [6].

In terms of the hotel sector, customers are active participants in the process of creating and developing new services and they contribute to innovation at the sector level [38]. However, the participation of frontline employees should not be disregarded [1], as not only do they deliver the service, but they also create it [12]. Innovation can therefore be presented as a mediating factor between customer orientation and hotel performance [17], which helps maintain the positive relationship between customer orientation and innovation, as presented by a series of authors [19,38,39]. Based on this, the following hypothesis was defined:

**Hypothesis 1 (H1).** *Customer orientation has a positive impact on hotel innovation.*

Competitor orientation includes the assessment of current competitors, as well as potential ones, through the identification of technologies which are capable of satisfying the present and future needs of customers [14]. There is therefore a need to determine the right time for the entry of new technologies or the implementation of changes in existing technologies [40]. Accordingly, it is relevant to know the competitor in order to be able to identify the best action plans to protect or improve the hotel's position [38,41]. At the hotel level, competitor orientation appears to be a stimulus for innovation [14,42]. Based on this, the following hypothesis was defined:

**Hypothesis 2 (H2).** *Competitor orientation has a positive impact on hotel innovation.*

Learning orientation focusses on creating and developing new perceptions to bring about a change in behaviours [38,43]. For a given hotel, learning orientation will continually improve and expand the skills and knowledge of the employees [4], enabling the development of new and better ways of interacting with customers and the learning of new technical and social skills [38]. Learning orientation positively influences the proactive and innovative capacity of hotel managers [2] and stimulates the development of new services and/or the introduction of improvements in existing ones [25,38], as well as value creation [44]. Based on this, the following hypothesis was developed:

**Hypothesis 3 (H3).** *Learning orientation has a positive impact on hotel innovation.*

Technological orientation is fundamental to the innovation process of a given hotel when it corresponds to the development of new technologies which are designed to meet the needs of current customers [26,28]. The exploitation of new customers implies the adoption of a more proactive approach which facilitates a better understanding of these customers' unidentified needs [3,43], as well as making it possible to obtain knowledge on how to win over customers and develop a closer relationship with them [26]. Accordingly, technological orientation enables the hotel to understand how to harness its capacity to produce new technologies and use its technological knowledge to answer customers' needs and requirements better, and also how to anticipate the customer satisfaction process [28,45]. In general, technological orientation drives development and the adoption of more modern and innovative technologies [45] and thus brings about a competitive advantage [14,27,31]. Some authors have debated the positive impact of technological orientation on innovation and the development of dynamic capabilities in companies with different activity plans. For this study, we opt to just focus the study on hotels, resulting in the following hypothesis:

**Hypothesis 4 (H4).** *Technological orientation has a positive impact on hotel innovation.*

Entrepreneurial orientation seeks new market opportunities [15] and involves greater proactivity with market opportunities, risk aversion, and sensitivity to innovations [45], as it reflects a continuous search for new business opportunities [32]. As such, entrepreneurial orientation is the tendency for a particular hotel to try to reach new customers, to look for new opportunities, and to retain current customers through a dynamic approach to the application of marketing practices and the ability to react to changes that occur in the environment [45]. As noted by [5,12,35] and others, entrepreneurship orientation has a positive impact on hotel innovation. Based on this, we develop the following hypothesis:

**Hypothesis 5 (H5).** *Entrepreneurial orientation has a positive impact on hotel innovation.*

Innovation strategies correspond to the ability to respond to changes in a flexible manner [45] and they represent the vision of transforming any innovation into a product or/and service which can guarantee the conquest of new customers and a competitive advantage. Thus, in the hotel sector, it is necessary to continuously develop new services [38], as the quality of the service provided is a determinant of business performance (revenues, profits, return on investment and market share) [46]. Research [38,47] has demonstrated the positive effect of hotel innovation on performance. Based on this, we developed the following hypothesis:

**Hypothesis 6 (H6).** *Hotel innovation has a positive impact on the performance of a hotel.*

*2.2. Control Variables*

Previous studies have shown that seniority, size, and category of the hotel can influence innovation. The seniority of a hotel is measured by the number of years it has been active and the size of a hotel reflects the number of employees, whereas the category of a hotel evaluates the number of stars that have been awarded to the hotel (ranging from one star to five stars) [33,38,48].

**3. Methods**

*3.1. Research Model and Hypotheses*

This study is of an explanatory nature and it studies the impact of strategic orientations on hotel innovation. The research model (Figure 1) was built based on the analysis of prior research in a review of the literature.

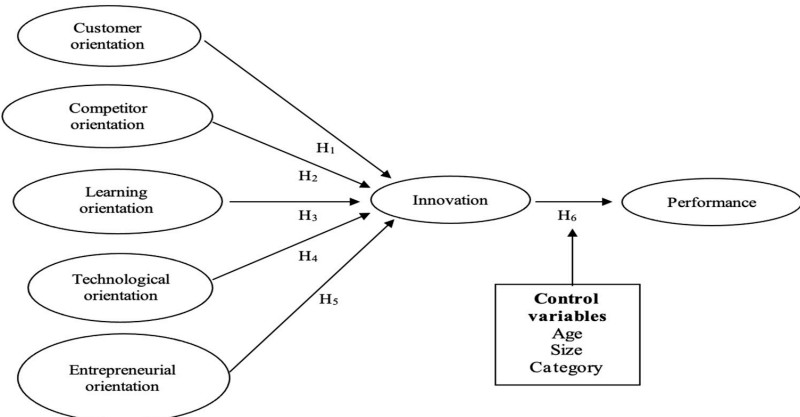

**Figure 1.** Research model and hypotheses.

### 3.2. Sample and Procedures

The population from which the sample was selected corresponds to hotels located in four provinces of Angola, with categories ranging from 1 to 5 stars. In view of the location, the four provinces with the highest profile in Angola were selected, namely: Luanda, Benguela, Huíla, and Namibe, with a total of 135 hotels (69%). A preliminary survey was prepared, including items of measures adopted in previous studies, such as a data collection tool. A non-probability purposive sampling technique was used to select a sample of 130 hotels. The data were collected through a self-administered survey, of which 71 questionnaires were filled out by the managers of the different hotels in Angola belonging to the four (4) provinces under study. Two questionnaires (3%) were excluded because they lack data, as recommended by [49].

In all, 69 questionnaires were considered usable for data analysis, which represented an effective response rate of 88.5%. A statistical power analysis was performed for sample size estimation. A test power (0.80), moderate effect size (0.35) [50], and five predictors of Innovation were used (see Figure 1). With an alpha = 0.05, the projected sample size needed with this effect size (G*Power 3.1) [50] is approximately 43 for an actual power = 0.80 and critical F = 2.46. The study sample size of hotels (N = 69) is adequate for the main objective of this research.

### 3.3. Measurement

To measure client orientation, we use the scale proposed by [13] and also used by [20,31], to reflect the level of knowledge about the client that can be collected to better understand their wishes and expectations. For competitor orientation, we use the scales developed by [13] and presented by [18,20], which reflect the level of information that hotels need to know about their competitors. For the learning orientation, we use the scales developed by [51] and used by [38,52] to reflect both the relevance of organizational learning in the development of new services and the introduction of improvements. For technological orientation, we use the scales developed by [28] and subsequently used by [53,54] to reflect the proactive capacity of a hotel to use cutting-edge technologies to bring about a better development of new products and/or services. For entrepreneurship orientation, we use the scales developed by [55] and used by [32,56]. For innovation we use the scales presented by [57]. Finally, to measure the hotels' performance, as this includes both financial and non-financial aspects [58], we use the scales used by [59,60], which were later developed by [13]. The five predictors and the two dependent variables were measured using a seven-point Likert scale, where 1 indicates "strongly disagree", and 7 indicates "strongly agree."

### 3.4. Data Analysis Tools and Techniques

Partial least square structural equation modeling (PLS-SEM) is frequently used for the analysis of non-normal data, as it is a strong and robust statistical tool which avoids

problems caused by small sample size [61–64]. Confirmatory factor analysis (CFA) was also used in this study, at two stages: first to evaluate the measurement model, and then to evaluate the structural model. The measurement model was used to evaluate the observed relationships between variables as well as the model's scales and the validity and reliability of the latent variables, while the structural model was used to investigate the relationships between those latent variables that had originally been determined to be reliable and valid.

## 4. Results

### 4.1. Respondents Profile

The main characteristics of the interviewees are that the majority are male (55.1%), aged between 25 and 35 years (47.8%). Most of the interviewees have a degree (78.3%) and perform the function of general manager in the hotel (45.0%). Most hotels have one star (27.0%) and have been in existence for less than 10 years. Lastly, most hotels have between 51 and 250 employees (36.3%). Table 1 summarises the demographic profile of the respondents.

**Table 1.** Demographic profile of respondents.

| Parameters/Categories | Frequency | % |
|---|---|---|
| Gender | | |
| Male | 38 | 55.1 |
| Female | 31 | 44.9 |
| Age Group | | |
| Less than 25 years | 9 | 13.0 |
| 25 to 35 | 33 | 47.8 |
| 36 to 45 | 16 | 23.2 |
| More than 45 | 11 | 16.0 |
| Education | | |
| High school or less | 15 | 21.7 |
| First Degree | 49 | 71.0 |
| Master's degree or higher | 5 | 7.3 |
| Responsibility | | |
| General Manager | 31 | 45.0 |
| Middle Manager | 19 | 27.5 |
| Line-level Supervisor | 13 | 18.8 |
| Administrative Staff | 6 | 8.7 |
| Hotel Years of Existence | | |
| Less than 10 years | 44 | 63.8 |
| 10 to 19 years | 22 | 31.9 |
| 20 to 30 years | 1 | 1.4 |
| More than 30 years | 2 | 2.9 |
| Hotel Category | | |
| One star | 19 | 27.0 |
| Two stars | 15 | 22.0 |
| Three stars | 13 | 19.0 |
| Four stars | 13 | 19.0 |
| Five stars | 9 | 13.0 |
| Hotel Size (number of employees) | | |
| Less than 11 | 19 | 27.5 |
| 11 to 50 | 21 | 30.4 |
| 51 to 250 | 25 | 36.3 |
| More than 250 | 4 | 5.8 |

Sample: N = 69.

### 4.2. Measurement Model Evaluation

The standardized root mean square residual (SRMR) is a measure of the research model's approximate fit, with a lower SRMR indicating a better fit. The measurement model produced an SRMR of 0.05, which was lower than the 0.08 threshold suggested by [65]. In the PLS-SEM, the evaluation of the internal consistency of reliability and validity of the measurements is the first criterion to be used to evaluate a reflective measurement model [61–63]. Table 2 shows the evaluation of construct validity, where the factor load for each item was higher than the recommended limit of 0.70 [61], which shows that all the indicators were convergent and valid. In addition, the AVE statistics for each of the constructs varied between 0.4 and 0.7 and all the indicators have statistically-significant external loads and are above the cut-off value of 0.4. Six of the indicators recorded values between the range of 0.4 and 0.7, namely CO2, TO1, TO2, EO1, EO2, EO5 and P7. However, these indicators have not been eliminated, as this would affect the validity of the construction content, with no resultant significant impact on compound reliability: except for P7, as this was well below 0.40, and EO1, which registered an AVE value below 0.50. After eliminating both P7 and OE1, Cronbach's Alpha and the composite reliability became 0.85 and 0.89 for P, and 0.73 and 0.83 for EO. After these adjustments, the scales considered in the model demonstrate the good reliability of the indicator.

**Table 2.** Evaluation of Construct Validity.

| Construct Items | Convergent Validity | | Construct Reliability | |
| --- | --- | --- | --- | --- |
| | Loadings | AVE | $\alpha$ | CR |
| Customer Orientation (CO) (Source: [13,20,31]) | | 0.58 | 0.85 | 0.89 |
| CO1—Our hotel believes in total commitment to the customer | 0.856 | | | |
| CO2—Our compensation plan compensates employees and managers are committed to customer satisfaction | 0.673 | | | |
| CO3—Customer satisfaction is regularly measured | 0.745 | | | |
| CO4—We make a great effort to meet customer needs better | 0.797 | | | |
| CO5—We do whatever it takes to create greater value for our customers | 0.750 | | | |
| CO6—We continuously monitor the needs of our customers | 0.722 | | | |
| Competitor Orientation (CRO) (Source: [13,18,20]) | | 0.60 | 0.79 | 0.86 |
| CRO1—The hotel responds quickly to competitive threats | 0.729 | | | |
| CRO2—Frontline employees regularly share information about competitors' strategies internally | 0.933 | | | |
| CRO3—Top management regularly discusses competitors' strengths and strategies | 0.768 | | | |
| CRO4—For the hotel, the target customers represent an opportunity to achieve a competitive advantage | 0.730 | | | |
| Learning Orientation (LO) (Source: [38,51,52]) | | 0.63 | 0.80 | 0.87 |
| LO1—At our hotel, employee training is an investment, not an expense | 0.729 | | | |
| LO2—The basic values of the service marketing system include learning as a key to improvement | 0.933 | | | |
| LO3—When the hotel stops learning from the marketing process, it poses a risk to the future | 0.768 | | | |
| LO4—Learning capacity is the key to improving the sales process of the service | 0.730 | | | |
| Technological Orientation (TO) (Source: [28,53,54]) | | 0.53 | 0.70 | 0.81 |
| TO1—The hotel uses sophisticated technologies to develop new products and/or services | 0.625 | | | |
| TO2—The new services developed always use the latest technology | 0.618 | | | |
| TO3—The hotel accepts technological innovation which is based on the results of market research | 0.833 | | | |
| TO4—The hotel immediately accepts any technological innovation which is developed by the program and/or project management of new products and/or services | 0.797 | | | |

**Table 2.** *Cont.*

| Construct Items | Convergent Validity | | Construct Reliability | |
| --- | --- | --- | --- | --- |
| | Loadings | AVE | α | CR |
| Entrepreneurial Orientation (EO) (Source: [32,55,56]) | | 0.54 | 0.73 | 0.83 |
| EO2—To deal with competitors, we are rarely the first business to introduce new products and/or services, administrative techniques, technologies, etc. | 0.670 | | | |
| EO3—In general, key managers have a strong tendency to follow the market leader in introducing new products or new ideas | 0.837 | | | |
| EO4—My hotel is aggressive and fiercely competitive | 0.743 | | | |
| EO5—To deal with competitors, we usually avoid competitive confrontations, preferring to adopt a "live and let live" posture | 0.692 | | | |
| Performance (P) (Source: [13,59,60]) | | 0.53 | 0.85 | 0.89 |
| P1—Our customer is loyal | 0.633 | | | |
| P2—Our customer is satisfied | 0.723 | | | |
| P3—Our products and/or services bring value to the customer's life | 0.817 | | | |
| P4—Our client is willing to be retained by the hotel | 0.766 | | | |
| P5—The hotel's market share is growing | 0.784 | | | |
| P6—Hotel sales are growing | 0.755 | | | |
| P8—The hotel's return on investment (ROI) is growing | 0.608 | | | |
| Innovation (I) (Source: [57]) | | | | |
| I1—The products and/or services developed by the hotel are very creative | 0.851 | | | |
| I2—The products and/or services designed by the hotel are often seen as being new to the market | 0.804 | | | |
| I3—The products and/or services developed have a strong impact on the hotel | 0.864 | | | |
| I4—The products and/or services developed by the hotel often involve the use of new techniques | 0.777 | | | |

Notes: AVE = Average variance extracted; α = Cronbach's alpha; CR = Composite reliability.

All constructs have an AVE with values equal to or greater than 0.50, which demonstrates that there is support to guarantee validity and shows adequate convergent validity for all constructs. To examine the discriminant validity, the criterion used was the one which represents the ratio of the correlations between traits to the correlations within traits. In PLS-SEM, this criterion is recommended to assess the discriminant validity [61,63]. Table 3 shows that all values shown diagonally are higher than any of the squared correlations of the other constructs [66]. Accordingly, overall, the constructs considered in the model demonstrate good discriminant validity.

**Table 3.** Evaluation of Discriminant Validity.

| | P | I | LO | CO | CRO | EO | TO |
| --- | --- | --- | --- | --- | --- | --- | --- |
| **P** | **0.730 ***** | | | | | | |
| **I** | 0.701 *** | **0.825 ***** | | | | | |
| **LO** | 0.538 *** | 0.468 *** | **0.794 ***** | | | | |
| **CO** | 0.519 *** | 0.300 ** | 0.650 *** | **0.759 ***** | | | |
| **CRO** | 0.384 *** | 0.354 *** | 0.485 *** | 0.433 *** | **0.777 ***** | | |
| **EO** | 0.331 *** | 0.450 *** | 0.256 ** | 0.185 * | 0.411 *** | **0.737 ***** | |
| **TO** | 0.382 *** | 0.432 *** | 0.480 *** | 0.492 *** | 0.406 *** | 0.364 *** | **0.725 ***** |

* $p < 0.10$; ** $p < 0.05$; *** $p < 0.01$. P: Performance, I: Hotel Innovation, LO: Learning Orientation, CO: Customer Orientation, CRO: Competitor Orientation, EO: Entrepreneurial Orientation, TO: Technological Orientation. The square root of AVE is shown diagonally in bold.

### 4.3. Structural Model Evaluation and Hypothesis Testing

In PLS-SEM, the strength of the path coefficients, the value of $R^2$ (coefficient of determination), and $f^2$ (the effect size) are the main criteria for the evaluation of the structural model [61,63]. Therefore, as a first criterion, the results of the path coefficient of the structural model to support the proposed hypotheses are presented in Table 4 (Figure 2).

The results indicate that the effect of customer orientation on innovation (H1: β = −0.080; t = 0.429; *p* > 0.05) is negative and is not statistically significant. The effect of competitor orientation on innovation (H2: β = 0.024; t = 0.190; *p* > 0.05) and the effect of technological orientation on innovation (H4: β = 0.189; t = 1.494; *p* > 0.05) are both positive but not statistically significant. However, the effect of learning orientation on innovation (H3: β = 0.341; t = 2.343; *p* < 0.05), of entrepreneurship orientation on innovation (H5: β = 0.299; t = 3.054; *p* < 0.01), and innovation on performance (H6: β = 0.701; t = 10.640; *p* < 0.001) is positive in these three cases and the proposed hypotheses are thus supported as they are statistically significant.

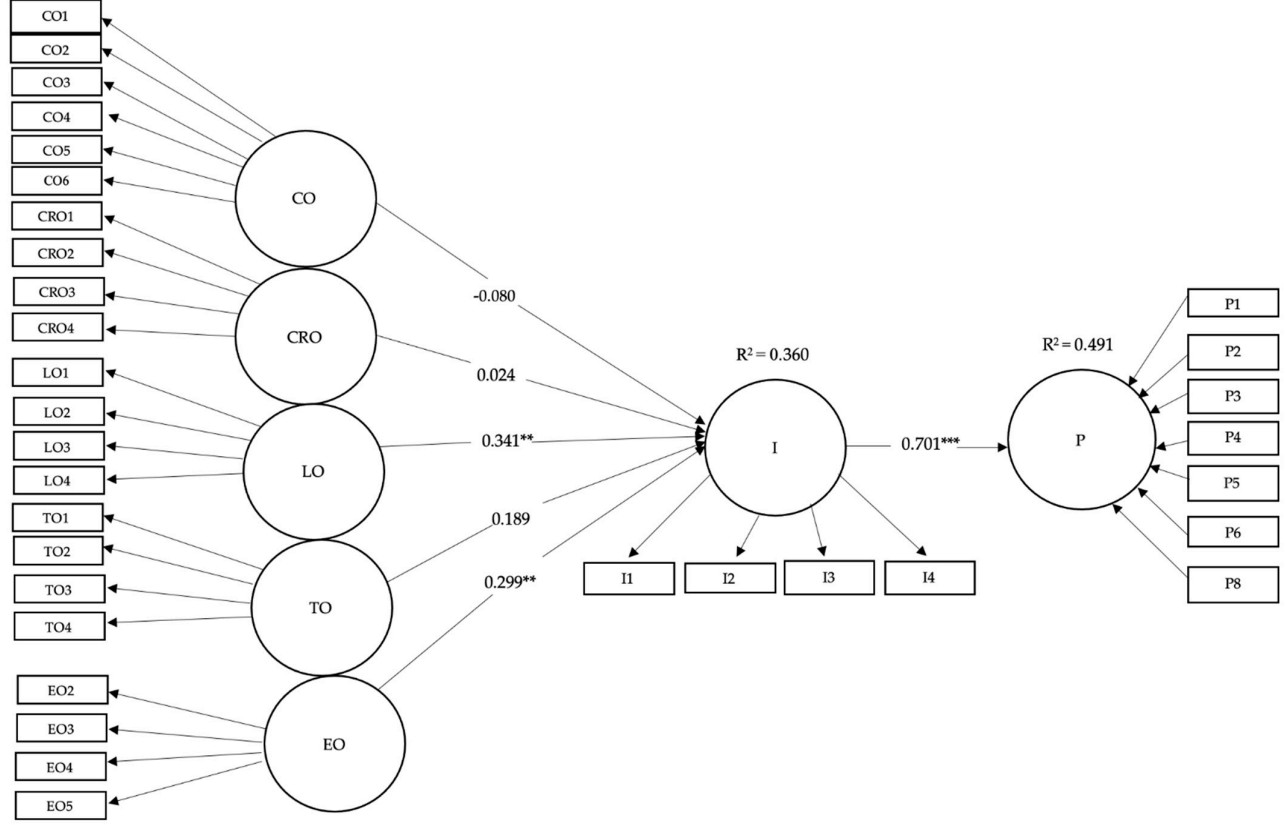

** *p* < 0.05; *** *p* < 0.01.
P: Performance, I: Innovation, LO: Learning Orientation, CO: Customer Orientation, CRO: Competitor Orientation, EO: Entrepreneurial Orientation, TO: Technological Orientation

**Figure 2.** Results of the Structural Model.

**Table 4.** Summary of the Hypotheses Testing.

| Hypotheses | Relationship | Estimate | SE | *t*-Statistics | Decision | f² (Effect Size) | q² | R² |
|---|---|---|---|---|---|---|---|---|
| H1 | CO→I | −0.080 | 0.186 | 0.429 | Insignificant | 0.002 | −0.002 | 0.360 |
| H2 | CRO→I | 0.024 | 0.124 | 0.190 | Insignificant | 0.000 | −0.009 | |
| H3 | LO→I | 0.341 | 0.146 | 2.343 ** | Significant | 0.058 | 0.032 | |
| H4 | TO→I | 0.189 | 0.127 | 1.494 | Insignificant | 0.023 | 0.015 | |
| H5 | EO→I | 0.299 | 0.098 | 3.054 *** | Significant | 0.065 | 0.036 | |
| H6 | I→P | 0.701 | 0.066 | 10.640 *** | Significant | | | 0.491 |

** *p* < 0.05; *** *p* < 0.01.

Table 4 shows the size effect (f²) for each path coefficient used to assess the extent to which the predictor variables affect the dependent variable. In PLS, the size of the f²

effect is measured using the PLS algorithm, in which the $f^2$ values of 0.02, 0.15, and 0.35 are respectively considered to be a weak, moderate, or strong effect on the relationship between the independent variable and dependent [61]. Customer orientation ($f^2$ = 0.002) and competitor orientation ($f^2$ = 0.000) display weak size effects on innovation. On the other hand, learning orientation ($f^2$ = 0.058), technological orientation ($f^2$ = 0.023), and entrepreneurship orientation ($f^2$ = 0.065) show moderate size effects on innovation. The indicators which have predictive relevance ($q^2$) are learning orientation, technological orientation, and entrepreneurship orientation, with values of 0.032, 0.015, and 0.036, respectively.

### 4.4. Post-Hoc Analysis: Categorical Moderators

In addition to testing the six hypotheses, we assessed the impact of strategic orientations on innovation and performance more closely in order to investigate whether there are differences for the following three moderators [67]: hotel years of existence, hotel category, and hotel size. To evaluate the moderating effect of years of existence of the hotel two groups were created: Group A = hotels less than 10 years old (N = 44), and Group B = hotels 10 or more years old (N = 25). Two groups have been established for classifying hotels according to quality: Group A = 2-star or less hotels (N = 34), and Group B = 3-star or greater hotels (N = 35). The moderating impact of the size of the hotel was measured by two groups: Group A = hotels with less than 50 employees (N = 19), and Group B = hotels with a minimum of 50 employees (N = 50). The moderating effect of a categorial moderator was assessed with SmartPLS, using PLS-MGA analysis. Table 5 shows that hotels with ten or more years of existence are less innovative than more modern ones. In terms of hotel category, it is notable that in high quality hotels innovation has a stronger impact on the hotel's performance. On the contrary, hotels with a larger number of employees seem as innovative as those with less employees. The conclusions found are in line with the results presented by [2,12,38].

**Table 5.** Group Difference Estimates.

| Moderator/Path | Path Coefficients | | | A vs. B | |
|---|---|---|---|---|---|
| | **Group A** | **Group B** | **A–B** | | |
| Years or existence | <10 (N = 44) | >=10 (N = 25) | Diff. | *t*-Statistic | *p*-Value |
| CO→I | 0.049 | −0.584 | 0.633 | 1.520 | 0.133 |
| CRO→I | 0.060 | 0.055 | 0.005 | 0.014 | 0.989 |
| LO→I | 0.164 | 0.682 | −0.518 | 1.409 | 0.163 |
| TO→I | 0.044 | 0.317 | −0.273 | 0.915 | 0.364 |
| EO→I | 0.480 | 0.340 | 0.140 | 0.481 | 0.632 |
| I→P | 0.689 | 0.772 | −0.083 | 0.708 | 0.481 |
| Category (stars) | <3 (34) | >=3 (35) | Diff. | *t*-Statistic | *p*-Value |
| CO→I | 0.175 | −0.326 | −0.610 | 1.228 | 0.224 |
| CRO→I | 0.039 | −0.052 | 0.090 | 0.300 | 0.765 |
| LO→I | 0.056 | 0.666 *** | −0.610 | 1.883 | 0.064 * |
| TO→I | 0.304 | 0.124 | 0.180 | 0.688 | 0.494 |
| EO→I | 0.276 | 0.355 ** | −0.078 | 0.333 | 0.740 |
| I→P | 0.571 *** | 0.844 *** | −0.273 | 2.185 | 0.032 ** |
| Size (employees) | <50 (19) | >=50 (50) | Diff. | *t*-Statistic | *p*-Value |
| CO→I | 0.526 | −0.072 | 0.599 | 1.196 | 0.236 |
| CRO→I | −0.089 | 0.006 | −0.095 | 0.297 | 0.767 |
| LO→I | −0.105 | 0.431 ** | −0.536 | 1.352 | 0.181 |
| TO→I | 0.301 | 0.100 | 0.201 | 0.635 | 0.528 |
| EO→I | 0.318 | 0.350 *** | −0.032 | 0.118 | 0.907 |
| I→P | 0.473 | 0.812 *** | −0.339 | 1.180 | 0.242 |

* $p < 0.10$; ** $p < 0.05$; *** $p < 0.01$. P: Performance, I: Innovation, LO: Learning Orientation, CO: Customer Orientation, CRO: Competitor Orientation, EO: Entrepreneurial Orientation, TO: Technological Orientation.

## 5. Discussion and Recommendation

### 5.1. Discussion

This study aims to assess the impact of strategic orientations on the innovation and performance of hotels in Angola. This multitheoretical research considers the theory of customer orientation, competitor orientation, learning orientation, technological orientation, entrepreneurship orientation, innovation, and hotel performance. The results obtained by PLS-SEM showed that only learning orientation (H3) and entrepreneurship orientation (H5) imply innovation at the hotel level. A strong learning orientation (H3) at the hotel level requires gathering a set of information about customers and competitors (direct or indirect), as well collecting information regarding political, economic, sociocultural, and technological changes in the hotel market, according to [1]. Based on this set of relevant information, employees' willingness and desire to learn more about new market trends and increases in customer feedback stimulate the hotel to absorb the benefits of the information collected, which subsequently leads to a trend towards innovation. Therefore, learning is a foundation for innovation according to [2,3,25,38,68,69], and learning shows a strong capacity for learning-oriented organisational strategy when applied to hotels, which has a large effect on innovation, as observed in our study.

The adoption of a strategy geared towards entrepreneurship orientation (H5) encourages the hotel to create new opportunities within existing products and services, or/and to renew those that do not add value, as suggested by [3]. Although the life cycle of the innovations developed at the hotel level is short, entrepreneurial capacity is fundamental to stimulate the constant implementation of successful innovations—which are generally those that occur when managers recognise a certain discrepancy between the needs of customers and the hotel's offer—and the successful attribution of the necessary resources to support these innovations. It should not be overlooked that entrepreneurship orientation has a positive relationship on the innovation of the hotel as well as on the performance of the same, due to the proactivity and the competitive capacity, as presented by [9,12,32–34,56,70] and others. Based on the multi group analysis, we observed that high quality hotels display even higher effects of innovation on performance, which is aligned with the results of [24].

### 5.2. Limitations

This analysis was restricted to the evaluation of the relationships between the variables in a given period, which somewhat reduces the explanatory capacity of the variables, following [1,5]. The second limitation is the fact that this study uses subjective criteria to assess endogenous variables, which makes concrete measurement and obtaining more accurate information difficult, according to [12]. A third limitation lies in the fact that sociodemographic variables, such as the age of the respondent, gender, or qualifications were not used to explore hotel innovation in more depth, as suggested by [39]. Even though PLS-SEM produces robust results for small samples, the study would have benefitted from a larger sample size [31], which would have made the analysis more conclusive.

### 5.3. Suggestions for Future Research

Future research should seek to deepen the understanding of the effects of strategic orientations on innovation, through the incorporation into hypothetical models of other constructs that prove to be appropriate and relevant. As suggested by [1,31], carrying out a longitudinal study would increase the explanatory capacity of the variables, due to the increase of the length of the data collection period, which would thus help identify the direction of existing causality between the variables. The carrying out of mixed, quantitative, and qualitative research is ideal for small samples. Future studies would do well to consider carrying out cross-cultural research, combining different countries, different incidents, and financial crises with different opinions and perceptions regarding innovation [18,39].

## 6. Conclusions and Implications

The results of this research on the impact of strategic orientations on innovation and hotel performance in Angola are consistent with those of other studies for other countries. Accordingly, this study has contributed a series of theoretical and practical implications for research on innovation in the hotel sector.

### 6.1. Theoretical Implications

New research on the impact of innovation on performance at the hotel level has emerged during recent years, driven by the main objective of promoting the creation of solid frameworks to assist hotel managers achieve the highest level of performance, through a series of strategic orientations, for example [3,12,13,28,31,35,67,71–74]. However, little effort was made to develop an integrative approach regarding hotel innovation, through identifying the relevant strategic orientations for the success of a hotel, especially in developing countries. This research thus contributes to the debate on the above-mentioned issues. In the first place, this research improves the understanding of the innovation of products and services, by presenting an integrative perspective in response to the academic call presented by several researchers, for example [1,2,12,35,75–77]). The results obtained show strong evidence for the applicability of an integrating model for hotel innovation. Secondly, this research provides a series of fundamental conditions for achieving a better understanding of innovation and its background through the hierarchy, which demonstrates a strong basis for the decision-making process. Quantitative evidence supports that learning orientation and entrepreneurship orientation are drivers for innovation, which in conjunction lead to improved performance. It is also important to highlight the relevance of organisational learning and the capacity for entrepreneurial learning in the process of hotel innovation in Angola as a developing country.

Finally, this research enabled the identification of a causal system of relationships and a better understanding of hotel innovation and performance.

### 6.2. Management Implications

The contents covered in this research contributed by benefitting the hotel industry in developing countries in general, and Angolan hotels in particular. Specifically, this research encourages hotel managers to adopt differentiated strategies designed to develop more creative innovation processes which will enable them to achieve improved performance. To this end, it is essential that hotel managers are aware of the benefits of innovation in services and products (for example, being open to new ideas, the tendency to generate new ideas by both internal and external stakeholders, and the creation of different portfolios) for gaining a competitive advantage.

Furthermore, hotel managers need to introduce a visible innovation process that focuses on objectives, goals, and strategies, which implies the creation of a multifunctional, communicative, and integrated innovation team, involving important functional areas such as Research & Development, Marketing, Technical Assistance, and Sales. This study shows the benefits of information sharing process, favouring the adoption of a strategy which is oriented towards learning and is based on the development of an increasingly attractive portfolio which stimulates the implementation of an entrepreneurship-oriented strategy [56]. This approach would enable hotels to benefit from a communication strategy based on coordinated and systemic communication involving all members of the organisation, with the objective of improving the innovation process, whilst facilitating the decentralisation of the decision-making process and greater autonomy among employees, together with encouraging the existence of more innovative behaviours and the ongoing search for opportunities and entrepreneurship. Accordingly, hotel managers are encouraged to create a culture of continuous learning to better strengthen the innovative capabilities of their personnel. Moderators related to years of existence, number of employees, and hotel category are not sufficient to differentiate hotels in terms of innovation.

**Author Contributions:** G.H.S. and J.M.C.V. contributed equally to this paper. They conducted the conceptualization, writing, formal analysis and validation and data curation. Both authors have read and agreed to the published version of the manuscript.

**Funding:** The study presented in this paper is funded by FCT—Fundação para a Ciência e Tecnologia (Portugal) national funding through research grant UID/SOC/04521/2019.

**Institutional Review Board Statement:** Not applicable.

**Informed Consent Statement:** Not applicable.

**Acknowledgments:** We would like to thank all the participants of the study.

**Conflicts of Interest:** Authors declare that there is no conflict of interest.

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
