# Peer review of "The Relationship between Strategic Orientation, Service Innovation, and Performance in Hotels in Angola"

_sustainability, doi:10.3390/su13116256_

Round 1
Reviewer 1 Report
Thank you for the opportunity to review this work. I admit that this is a quite well-prepared work and the authors' experience is noticed, although a few remarks can be made that may help improve this publication.
- The introduction to this work lacks a goal (including it in the abstract and methodology is not enough). What goals (cognitive, methodical, utilitarian, ...) do the authors want to achieve with this article?
- The research methodology is not clearly described. A non-probabilistic sampling technique was used to select a sample of hotels. But what was it about? Please also discuss the research questionnaire carefully. I am glad that the authors refer to the scales used by other researchers, but they need to be described, not just indicate the source. Should the reader look for them in other publications?
-
In the theoretical part, the authors decided to successively formulate hypotheses. Perhaps it is a convenient form of presentation for them, but at the same time unusual. I am not saying that this is a mistake, but I have the impression that I am reading an abbreviation of a scientific monograph and not an article.
- I also miss explain the basic terms used by the authors, especially those used in the subject line.
- The formulation of hypotheses is very limited and is based on proving a positive impact on hotel innovation. Neither hypothesis has an alternative hypothesis. Why? This suggests that they are obvious and raises questions about the meaning of this research.
The article requires improvement and re-verification, but I wish the authors perseverance and look forward to a revised version of their scientific work.
Author Response
Dear Reviewer,
The authors thank you for your comments and suggestions, which are addressed below. We took this opportunity to improve the flow of the text and expand the post-hoc analysis. The paper was reviewed with track changes so that changes are easily visible and was proofread by a native English speaker.
Please see the attachment.
Regards

Reviewer 2 Report
This study examines the relationship between strategic orientation, service innovation, and performance in hotels in Angola. Thank you for giving me opportunities to review this manuscript. I have some comments based on specially the small sample and the results.
Abstract
The abstract needs to summarize the purpose, data, method, results and implications in brief. The abstract emphasizes the lack of case studies in Angola. The abstract needs to be revised.
Introduction
On page 2 line 39
The sentence seems to be not completed. (a positive impact on the company’s performance, Chou et al,).
Introduction
The antecedents of the proposed model are well explained, however, introduction should present the problem statement, the theoretical framework, and purpose of this study. I am not sure why this study is necessary and important and why this study should be conducted for the hotel industry.
Data
The study collected data from 69 respondents. This number is not sufficient to examine the proposed model by using structural equation modeling analysis.
The authors should present the power of the sample because the variables in the proposed model are too many to have robust results with the 69 samples.
Results
Demographic information should be presented.
Results
The model fit results of confirmatory factor analysis and Structural equation modeling are not presented.
Results
The innovative items appeal to the innovative technology, however, the results show the relationship between technological orientation and innovation was not statistically significant. The results should provide sufficient explanations.
Results
In Table 1, one of the items (D7 Hotel sales are growing) is 0.134. The factor loading is too small.
Implications
The sentence below in 6.1. may not be appropriate because the results were not all statistically significant.
“Quantitative evidence supports that certain strategic orientations (customer orientation, competitor orientation, learning orientation, technological orientation, and entrepreneurship orientation) are shown to be excellent drivers for innovation.”
Author Response

(The authors gave the same response as above.)

Round 2
Reviewer 1 Report
Thank you for improving the article. I have no objections to the current version of the article.
Congratulations and good luck
Reviewer 2 Report
Thank you for giving me an opportunity to review this paper.
I still believe the number of 69 respondents are inappropriate to confirm the proposed conceptual model in this paper. Even though the authors present numbers in tables, the results are not reliable.
The explanation about the power analysis is not appropriate. The authors insisted that there are five predictors. The authors should ensure the readers about the power and how they calculated the effect and the appropriate sample number. I recommend the authors refer to Kline (2015) and check out the sample number.
This paper did not present the model fit of CFA and SEM at all.